# A Secure Blockchain-Based Authentication and Key Agreement Scheme for 3GPP 5G Networks

**DOI:** 10.3390/s22124525

**Published:** 2022-06-15

**Authors:** Man Chun Chow, Maode Ma

**Affiliations:** 1School of Electrical and Electronic Engineering, Nanyang Technological University, Singapore 639798, Singapore; manchun001@e.ntu.edu.sg; 2College of Engineering, Qatar University, Doha P.O. Box 2713, Qatar

**Keywords:** 5G, 5G-AKA, authentication, blockchain, BAN logic, Scyther

## Abstract

The futuristic fifth-generation cellular network (5G) not only supports high-speed internet, but must also connect a multitude of devices simultaneously without compromising network security. To ensure the security of the network, the Third Generation Partnership Project (3GPP) has standardized the 5G Authentication and Key Agreement (AKA) protocol for mutually authenticating user equipment (UE), base stations, and the core network. However, it has been found that 5G-AKA is vulnerable to many attacks, including linkability attacks, denial-of-service (DoS) attacks, and distributed denial-of-service (DDoS) attacks. To address these security issues and improve the robustness of the 5G network, in this paper, we introduce the Secure Blockchain-based Authentication and Key Agreement for 5G Networks (5GSBA). Using blockchain as a distributed database, our 5GSBA decentralizes authentication functions from a centralized server to all base stations. It can prevent single-point-of-failure and increase the difficulty of DDoS attacks. Moreover, to ensure the data in the blockchain cannot be used for device impersonation, our scheme employs the one-time secret hash function as the device secret key. Furthermore, our 5GSBA can protect device anonymity by mandating the encryption of device identities with Subscription Concealed Identifiers (SUCI). Linkability attacks are also prevented by deprecating the sequence number with Elliptic Curve Diffie–Hellman (ECDH). We use Burrows–Abadi–Needham (BAN) logic and the Scyther tool to formally verify our protocol. The security analysis shows that 5GSBA is superior to 5G-AKA in terms of perfect forward secrecy, device anonymity, and mutual Authentication and Key Agreement (AKA). Additionally, it effectively deters linkability attacks, replay attacks, and most importantly, DoS and DDoS attacks. Finally, the performance evaluation shows that 5GSBA is efficient for both UEs and base stations with reasonably low computational costs and energy consumption.

## 1. Introduction

In recent years, the exponential growth of mobile subscribers and smart devices has fostered the rapid development of the fifth-generation cellular network (5G). Unlike the conventional 4G networks that only support limited numbers and types of devices, the 5G network is designed to connect as many devices as possible within one network. All devices such as mobile phones, autonomous vehicles, and Internet of Things (IoT) can now connect to the same 5G network with optimal speed and latency. To cater to these stringent requirements, the 5G network is constructed by many tiny femtocells to serve many users [1]. In this way, limited spectrum resources can be reused effectively to provide services to more devices simultaneously. Additionally, having more base stations installed, 5G wireless networks can alleviate traffic congestion in wireless channels. Hence, the futuristic 5G networks improve wireless connections with faster speed, lower latency, and greater capacity.

Although the 5G network is said to provide numerous benefits, there are also many new security challenges. In view of these potential security issues, the Third Generation Partnership Project (3GPP) has standardized a new Authentication and Key Agreement (AKA) protocol known as 5G-AKA in TS 33.501 [2]. 5G-AKA can mutually authenticate base stations, 5G core networks, and user equipment (UE). It has resolved some pre-existing security issues found in the 4G Long-Term Evolution (LTE) networks. For example, by encrypting the permanent identity of the UE using the Subscription Concealed Identifier (SUCI), 5G-AKA can prevent International Mobile Subscriber Identification (IMSI), catching attacks and rogue base station attacks [3]. However, as some of the authentication methods in 5G-AKA are inherited from the 4G EPS-AKA [4], security issues in 4G networks remain unsolved in 5G-AKA. For example, 5G-AKA suffers from linkability attacks, in which malicious users can track a specific device by using synchronization error messages [5]. Additionally, it lacks perfect session key forward secrecy that guarantees data confidentiality, even if the long-term key is stolen in the future [3]. Furthermore, 5G-AKA is a centralized protocol that relies heavily on two functional entities, namely the Authentication Server Function (AUSF), and the Authentication credential Repository and Processing Function (ARPF) located inside the Unified Data Management (UDM) server. As there will be a tremendous number of devices connecting to the same 5G core network, 5G-AKA would be highly vulnerable to Denial of Service (DoS) attacks and Distributed Denial of Service (DDoS) attacks which aim to paralyze functional entities in the core network. Consequently, the existing 5G-AKA protocol is subject to many security threats that affect the robustness and reliability of the 5G network.

On the other hand, blockchain is a new technology for decentralized applications. Initially proposed by the creator of Bitcoin cryptocurrency, blockchain is a practical way to construct and manage a trustworthy decentralized ledger database across the network. By storing transactions into data blocks and linking them together using cryptographic hash functions, blockchain ensures all blocks in the chain reach the consensus effectively. Additionally, blockchain ensures the data in the database becomes computationally infeasible to mutate. In recent years, researchers have envisioned that blockchain can be used in a distributed way to solve many challenging problems in the 5G network [6]. As the number of 5G network infrastructures and mobile devices is growing exponentially, the benefits of blockchain would become more prominent in the future.

In this paper, we propose a Secure Blockchain-based Authentication and Key Agreement scheme for the 3GPP 5G network (5GSBA). Our 5GSBA protocol offers these benefits: first, it provides a distributed way to store all subscribers’ information safely. By employing the one-time hash secret, authentication-related entries are the hashed digests that work similarly as public keys. Hence, even if the database is disclosed in the future, adversaries cannot use it to impersonate any UEs. Moreover, 5GSBA prevents not only typical network attacks such as eavesdropping, man-in-the-middle attacks, replay attacks, and IMSI-catching attacks, but also prevents DoS and DDoS attacks effectively and provides perfect session key forward/backward secrecy. The main contributions of this paper are as follows:We design a novel Authentication and Key Agreement protocol for the 3GPP 5G network. 5GSBA works based on the improvement of the existing system architecture of the 5G core network. It can be easily adopted to the 3GPP access scenario, in which all UEs are connected to the home network via nearby gNBs;Our proposed 5GSBA protocol is secure and efficient. Using blockchains and other state-of-the-art cryptographic functions, 5GSBA can guarantee device unlinkability, mutual authentication, and data confidentiality with low computational and energy costs. Most importantly, not only can all typical network attacks be prevented, but DoS and DDoS attacks can be deterred;The security of the protocol is verified with BAN logic and the formal verification tool Scyther. The performance evaluation and simulations also demonstrate its resistivity to DoS and DDoS attacks.

The rest of the paper is organized as follows. Section 2 reviews the existing works on 5G authentication and some blockchain-based 5G applications. Section 3 introduces the system and security model of 5GSBA. Section 4 discusses the motivations for and the details of 5GSBA. Section 5 presents the work of security evaluation and Section 6 presents the work of performance evaluation with some simulation results under different attacks. Finally, a conclusion is drawn in Section 7.

## 2. Related Work

Recently, various solutions have been proposed to improve security in the Authentication and Key Agreement (AKA) process in the 5G network. In this section, we first discuss the security vulnerabilities in the existing 5G-AKA protocol. Then, we briefly review the major research work related to our work, including blockchain-related 5G authentication schemes and AKA schemes against DoS attacks.

### 2.1. Security Vulnerabilities in 5G-AKA

5G-AKA is the standardized Authentication and Key Agreement protocol in the latest 3GPP 5G security architecture TS 33.501 [2]. Evolved from the architecture of EPS-AKA in the LTE security architecture, 5G-AKA aims to ensure the authenticity between UE, the serving network, and the home network. However, some security vulnerabilities in the 5G-AKA have recently been disclosed, making it less secure than has been claimed. For example, Ref. [3,5] found that 5G-AKA suffers from linkability attacks, by which adversaries can use synchronization error messages (MAC_FAIL and SYNC_FAIL) to detect if the UE is currently located in a certain area. Additionally, 3GPP TR 33.846 [7] found that 5G-AKA fails to prevent denial-of-service (DoS) attacks because the 5GC has no way to justify if the SUCI is a replayed message. The 5G-AKA is a centralized protocol that heavily relies on the authentication functional entities of AUSF/UDM, so it could be vulnerable to Distributed DoS (DDoS) attacks and single-point-of-failure issues in the AUSF/UDM. Moreover, [4] found that 5G-AKA fails to provide perfect forward secrecy and post-compromise secrecy due to the use of the long-term symmetric keys and sequence numbers. In fact, according to the 3GPP TS 33.501 [2], the device anonymity protection of UE in 5G-AKA is also vulnerable. For example, network operators can opt out of the encryption in the Subscription Concealed Identifier (SUCI) that encrypts the Subscription Permanent Identifier (SUPI) of UE. It is also known as a “null-scheme”. Thus, the UE will send the cleartext of its SUPI through wireless channels, which could be dangerous for IMSI-catching attacks. In some emergent situations, UE also sends its SUPI directly to initiate authentication procedures. To conclude, 5G-AKA is vulnerable to many network attacks, including but not limited to linkability attacks, DoS attacks, and IMSI-catching attacks. The lack of perfect forward secrecy also makes 5G-AKA vulnerable to session data recovery if the long-term key (LTK) is compromised at any time.

### 2.2. Blockchain in 5G Authentication

Linking data blocks into a chain, blockchain technology is essentially a secure decentralized database solution that guarantees data immutability and practical consensus across multiple network nodes. There are three different types of blockchain platforms [8]: permission-less, permissioned, and consortium blockchains. Among all three types of blockchains, it is envisioned that private and consortium blockchains are the most suitable distributed solutions to solve the security challenges in 5G because of their high efficiency [6].

In recent years, some proposals combining blockchains with 5G authentication have surfaced. For example, Yang et al. [9] introduced the idea of a blockchain-based anonymous access (BAA) scheme that allows equipment manufacturers, network operators, and users to access the blockchain-based database and perform mutual authentication. However, there is no formally proved protocol presented in the proposal. Haddad et al. proposed a blockchain-based 5G authentication protocol based on a public blockchain in [10,11]. They suggested that all 5G access points (APs) can use the UE public keys listed in the blockchain to perform mutual authentication between the AP and the UE. However, this misses a mechanism for UE to retrieve the public keys of the surrounding APs. Xu et al. [12] proposed the use of redactable blockchains to store all subscriber’s information. The redactable blockchain provides key deletion and revocation functions. It is beneficial for network operators to protect the privacy of their users. However, the proposal lacks an authentication protocol for UEs and core networks to secure user data using the keys in the blockchain. Jia et al. [13] proposed a decentralized authentication scheme for 5G IoT devices. This protocol suggests that authentication entities in all domains can upload their device registration records to the same alliance blockchain. However, the protocol uses an identity-based cryptosystem. It introduces high computational overhead to mobile devices and edge servers, making them energy inefficient and prone to request flooding. Liu et al. [14] proposed an efficient authentication protocol based on 5G extensible authentication protocol (5G EAP-AKA’) and a private blockchain. However, the security functionality of the proposed scheme has not been formally analyzed. Moreover, the EAP framework could also introduce more signaling overhead than the existing 5G-AKA scheme.

Some recent solutions have been designed to accelerate handover authentication in 5G wireless networks by using blockchains [15,16,17,18,19]. While all of them are providing a fast way to share the secret keys among base stations, most of them did not discuss how to prevent DoS and DDoS attacks during the UE authentication phase in a fast and efficient way. As a result, this shows that most of the existing blockchain authentication works are incomplete, and almost all of them could not alleviate the threats of DoS and DDoS attacks effectively during the UE authentication. In other words, designing a blockchain-based authentication protocol that provides adequate attack prevention, is energy-efficient, and computationally fast at the same time is a challenging research work. Overall, Table 1 summarizes all recent blockchain-based 5G authentication schemes and their challenges.

### 2.3. AKA Schemes against DoS and DDoS Attacks

On the other hand, many solutions have been designed to alleviate DoS and DDoS attacks during the UE authentication. Some recent 5G AKA schemes aimed at preventing DoS and DDoS are presented in Table 2. For example, by allowing multiple devices to choose a group leader as an agent, many group-based authentication protocols such as [20,21] aim to relieve computational burden and signaling overheads while authenticating a mass of devices. Although they can alleviate DoS attacks by including secret keys and short delays in the authentication requests, adversaries can bypass the security mechanism by launching the attacks individually. Leu et al. [22] have proposed to construct an AUSF pool and add a mediator to monitor all AUSFs. Although this provides disruption-free 5G-AKA authentication, it cannot prevent DoS attacks effectively because it is uses 5G-AKA, which is the protocol vulnerable to DoS attacks. Additionally, the bandwidth in the AUSF pool is still finite and expensive, and thus needs more investment. Yan et al. [23] have proposed a lightweight and secure handover authentication scheme based on a prediction of the potential target gNB from all neighbor gNBs in the 5G wireless network. The proposal facilitates fast 5G handover authentication by using time-to-live (TTL) attributes and encrypting the next hop chaining counter (NCC) with the Chinese remainder theorem. There are also some other schemes, including [24,25,26,27,28], proposed recently to fix other security issues in EPS-AKA and 5G-AKA. However, all of them suffer from single-point-of-failure due to the centralized protocol designs. Thus, all the existing DoS attack prevention schemes cannot provide adequate DoS and DDoS attack prevention.

## 3. System and Security Model

### 3.1. System Model

Our system model follows the 3GPP 5G system architecture listed in TS 23.501 [29] and the security architecture listed in TS 33.501 [2]. We are adding some new features to the existing functional entities. Figure 1 shows the 3GPP 5G core network (5GC) consisting of many functional entities.

In the existing 3GPP 5GC system model, the Next Generation Node B (gNB) is the base station that directly communicates with the UE. All gNBs in the 5G network are connected to the nearby Access and Mobility Function (AMF) servers. During conventional 5G-AKA authentication, when UE is under the coverage of 3GPP access (i.e., under the signal coverage of gNBs), it should send an authentication request to gNBs. Then, the gNB forwards it to the AMF, and the AMF forwards it to the Authentication Server Function (AUSF) server. After that, AUSF fetches the device secret keys from the Unified Data Management (UDM) server to continue the subsequent authentication steps. Normally, one AUSF serves many AMFs across the 5G core network, and each AMF serves many gNBs nearby.

In our proposal, we follow the same system model with a decentralization of the authentication entities as follows. All gNBs in the 5G core network become the members of a private blockchain that stores subscribers’ information. AMF and AUSF should no longer need to forward authentication requests, but they can optionally be one of the members in the blockchain. UDM is the protected repository that stores private keys of the blockchain and other important secrets. For each authentication request, UE uses the one-time hash secret stored in the Universal Subscriber Identity Module (USIM, or commonly known as a SIM card) to initiate an authentication request. Then, the gNB hashes the received secret and compares it with the entries in the blockchain. Since only legitimate UE would have the original secret, the gNB can immediately grant or deny the UE’s authentication request without forwarding the request to AMF, AUSF, or UDM.

Considering that 5G devices can connect to the 5GC by many different approaches such as non-3GPP access, 3GPP access in the home network, and 3GPP access in a visiting network, designing a universal authentication protocol could be very complicated. In this work, we focus on the most common scenarios to simplify the designed authentication protocol. It is assumed that all UE uses 3GPP access to connect to the gNBs in the home network. These gNBs are connected to the 5GC. Additionally, the connections between gNBs and 5GC are secured by wired connections protected by IPSec tunnels. Thus, if a gNB can mutually authenticate with UE using the private key of the home network and information stored in the blockchain, it can be regarded that the UE has joined a legitimate 5GC network.

### 3.2. Security Model

Our security model is also displayed in Figure 1. It is assumed that the 5G wireless channels follow the Dolev-Yao model [30], which assumes that there could be some neighboring active and passive attackers. Passive attackers eavesdrop and interpret the messages sent from both UEs and gNBs. Then, they can analyze the intercepted data to figure out the messages sent from both parties. Active attackers not only eavesdrop on the wireless channels, but also modify the intercepted messages, replay them, or even fabricate new messages to impersonate legal UEs or gNBs to disrupt the network. All these attackers are labeled as “Dolev-Yao (DY) Attackers” in Figure 1. Moreover, although it is assumed that the communication channels within 5GC are trustworthy, some accidents or misconfiguration could happen to the functional entities. For example, information in the data repositories of the 5GC functional entities could be leaked in some rare cases. In this scenario, these passive attackers would try to use the exposed permanent keys to decrypt previous communication data. Additionally, it is possible that there is compromised botnet UE in the wireless network. UE could launch DoS and DDoS attacks at any scheduled time, attempting to flood or paralyze the authentication-related functional entities in the 5GC.

Due to all the assumptions above, a desired 5G authentication protocol should provide security functionalities including device anonymity, mutual authentication, secure data transmission, and session key perfect forward secrecy. Moreover, it should be able to prevent active attacks such as impersonation, linkability attacks [5], replay attacks, man-in-the-middle (MITM) attacks, and DoS attacks. For passive attacks, there could be eavesdropping and location tracking attacks. These attacks must also be deterred.

## 4. The Proposed 5GSBA Scheme

### 4.1. Motivation

The existing 5G authentication protocol, 5G-AKA, is a vulnerable protocol that creates a vast burden to the AUSF. Since there will be more connected devices in the future, attackers are likely to launch DoS and DDoS attacks to flood the AUSF and other 5G authentication entities. However, as there could also be some essential utilities using 5G, the 5G network has to be stable at all times. Therefore, to prevent DoS and DDoS attacks from paralyzing the 5G network, there is an urgent and critical need to design an authentication protocol that ensures no DoS and DDoS attacks can be successful. This protocol should work in a decentralized manner, such that it will not suffer from a single point of failure due to request flooding. Given all these constraints, we believe that an authentication protocol combined with a blockchain would be an ideal solution. Although a blockchain could introduce more overhead during database synchronizations, it helps reduce the opportunity of system overloading by decentralizing authentication tasks. In this paper, we propose a 5GSBA scheme that uses a blockchain to decentralize the subscription repository from UDM to all gNBs, such that authentication tasks can be decentralized to the gNBs. By doing so, we can prevent DoS and DDoS attacks from impacting the quality of service (QoS) of the entire 5G network.

### 4.2. Details of the 5GSBA

This section presents the Secure Blockchain-based Authentication and Key Agreement Protocol (5GSBA) in detail, which is a two-step protocol designed to mutually authenticate between UEs and gNBs in the 5G network. To satisfy all aforementioned security requirements, the 5GSBA combines the one-time hash function, Elliptic Curve Diffie–Hellman (ECDH), the Elliptic Curve Integrated Encryption Scheme (ECIES), and the keyed-hash message authentication code (HMAC) at different steps of the protocol. The 5GSBA uses a private blockchain network across all gNBs as a distributed subscriber data repository. Therefore, all gNBs connected to the 5GC may approve authentication requests from UE autonomously without taxing the AUSF and UDM. The 5GSBA has four phases: the system initialization phase, the USIM registration phase, the gNB broadcast phase, and the mutual authentication phase. All notations used in this paper are listed in Table 3, and Figure 2 shows a sequence diagram explaining different phases in 5GSBA.

**Phase 1—System Initialization:** Let p be the modulus, EFp be the elliptic curve over a finite field Fp, P be the generator point on EFp with an order *n*, and G be the generated subgroup which multiplies the generator point P. Additionally, we let the cryptographic hash function be H⊆ℤn*. Having the assumptions above, gNBs and AUSF run the following procedures:AUSF generates a new ECIES private key SKcore representing the 5GC by choosing a random input k. Then, the ECIES public key PKcore=k·P is stored at UDM.When there is a new gNB joining the 5GC, they should mutually authenticate with any existing approach such as IPSec tunnels. After that, AUSF installs the ECIES private key SKcore from UDM into the secure enclave of the newly joined gNB.Finally, the authenticated gNB downloads the latest private blockchain from the 5GC. gNB may also index the transactions in the blockchain locally for faster access.Whenever the blockchain has any updates, the gNB will download and index the new blocks accordingly.

Regarding the private blockchain administered by AUSF, Figure 3 illustrates the structure of a block in the whole blockchain. Every block should contain the following elements:**Block Header:** It contains the block version for future maintenance and upgrades.**Previous Block Hash:** It is the hash of the previous block. It guarantees the immutability of the blockchain.**Timestamp:** It is the block creation time for tracking purposes. All transaction timestamps within the block should never be larger than this timestamp.**Transactions:** Each transaction is a subscription record for one device. To reduce the storage overhead of the blockchain, each block contains multiple transaction records. This size should be adjustable according to the preference of network operators. By default, we follow the block size of bitcoin as 1 MB.

Since it is a private blockchain, any efficient algorithm can be used, such as Practical Byzantine Fault Tolerance (PBFT) [31], to reach a consensus for all gNBs. The details about blockchain consensus implementation are omitted in this paper.

**Phase 2—USIM Production:** Network operators should install a one-time hash secret Y and the ECIES public key of the 5GC $PK_{core}$ to the USIM during USIM production. Additionally, the one-time secret hash digest $H\left(Y\right)$ should be posted to the private blockchain. In this way, when the UE sends the collision of the hash function (i.e., the secret Y) to the gNB, it can prove to the gNB that it is the legitimate UE. These are the detailed procedures:The operator generates a one-time hash secret Y and the digest of the one-time hash secret $H\left(Y\right)$;USIM stores its permanent identity (i.e., SUPI), elliptic curve parameters, one-time hash secret Y, and the ECIES public key of 5GC $PK_{core}$ into its non-volatile storage;AUSF creates a new blockchain transaction including the SUPI, $H\left(Y\right)$, timestamp, and a status code. The status code is the activation status of the SUPI. For example, “activated” can be 1, “suspended” can be 2, “revoked” can be 3, and so on. The format of one transaction in the blockchain is also shown in Figure 3.If we need to revoke the access of a specific USIM, AUSF can post a new transaction with a “revoked” activation status code and a timestamp to the blockchain. Therefore, when gNBs retrieve the latest transactions from the blockchain, they will follow the last record to deny access from that USIM.

**Phase 3—gNB Broadcast:** after initialization, gNBs broadcast their identities $ID_{gNB}$ through the air. Since 5GSBA makes SUCI mandatory during UE authentication, the identity request procedures in 3GPP TS 33.501 [2] are no longer needed. Hence, UE can freely choose when to start authentication and when to prepare SUCI without having to respond to possibly forged identity requests from gNBs.

**Phase 4—Mutual Authentication:** whenever UE is powered up to start authentication with the 5G network, the following two-step protocol will be executed:**UE****→ gNB:** UE sends an authentication request to the gNB with these steps:Generate a new random HMAC key $K_{hmac}$, random ECDH public key a⋅P, and timestamp TS;Generate SUCI by encrypting {SUPI, Y, $K_{hmac}$} with $PK_{core}$;Generate the next one-time hash secret $Y_{2}$, and calculate its hash $H(Y_{2})$;Update the Y in the local storage as $Y_{2}$, as it will become the Y for the next authentication;Calculate σ1 = HMAC ({$ID_{gNB},\;SUPI,\;Y,\;H\left(Y_{2}\right),\;TS,\;a\cdot P$} $K_{hmac}$);Send the authentication request = $\left\{SUCI,\;H\left(Y_{2}\right),\;TS,\;a\cdot P,\;\sigma 1\right\}$ to gNB.**gNB** checks the incoming authentication request with these steps:Check the validity of the timestamp TS, and then decrypt the SUCI into {SUPI, Y, $K_{hmac}$} using the private key $SK_{core}$;Verify the HMAC of the message σ1 = HMAC ({$ID_{gNB},\;SUPI,\;Y,\;H\left(Y_{2}\right),\;TS,\;a\cdot P$} $K_{hmac}$);Fetch the latest transaction of the SUPI from the private blockchain locally;Compare the hash of the received Y with the $H\left(Y\right)\;$value stored in the blockchain. If there is a collision (i.e., two values are equal), send an authentication response. Otherwise, gNB should stop the protocol;Create a new blockchain transaction containing the value of $H(Y_{2})$, and upload the block containing this transaction when the gNB is idle.**gNB****→ UE:** gNB issues an authentication response to the UE with these steps:Generate a new random ECDH public key b⋅P;Calculate σ2 = HMAC ({SUPI, TS, b⋅P}, $K_{hmac}$), where TS is the received timestamp;Send the authentication response = {TS, b⋅P, σ}.**UE** checks the incoming authentication response with these steps:Calculate σ’ = HMAC ({SUPI, TS, b⋅P}, $K_{hmac}$). If σ equals to σ’, the UE continues to calculate the common ECDH session key using formula a⋅b⋅P;Similarly, gNB calculates the common ECDH session key using formula b⋅a⋅P. Since a⋅b⋅P=b⋅a⋅P, a common session key is derived. Both parties are now mutually authenticated.

## 5. Security Evaluation

In this section, we firstly justify the logical correctness of the 5GSBA using Burrows–Abadi–Needham (BAN) logic. Then, we provide formal verification on the security of the 5GSBA using the Scyther formal verification tool. Moreover, we present an extensive qualitative security analysis based on the discussion in Section 3.2 to show that the 5GSBA is secure to fight against various malicious attacks.

### 5.1. Burrows–Abadi–Needham (BAN) Logic

Burrows–Abadi–Needham (BAN) logic is a set of logic rules to verify the logical correctness of an authentication protocol [32]. Assuming that the cryptographic functions in the protocol are perfect, BAN logic can systematically find out all incorrect designs in an authentication protocol. To apply BAN logic to our 5GSBA protocol, we formalize our protocol into the idealized form. Then, we use BAN logic symbols and rules [32] such as the message meaning rule, belief rule, nonce verification rule, jurisdiction rule, etc., to validate if our 5GSBA protocol fulfills the targeted security goals.

#### 5.1.1. Formalized 5GSBA Protocol

In our idealized protocol, U refers to UE and C refers to one of the gNBs in the 5G cellular network (CN). All cleartext and identities in the protocol are omitted as they can be easily forged. For the notations, SUCI can be regarded as a message encrypted by the public key of 5GC (i.e., the $PK_{core}$). The timestamp token is represented by TS, and the one-time hash token is represented by Y. In addition, all message content protected by the HMAC can be viewed as a message encrypted by the HMAC key (i.e., $K_{hmac}$). Therefore, the idealized 5GSBA protocol is shown below:

Message 1: $U\to C:\;C⊲{\left\{U\overset{K_{hmac}}{↔}C,\;Y_{Y}\right\}}_{PK_{core}},{\left\{\;TS,\;a\cdot G\;\right\}}_{K_{hmac}}$

Message 2: $C\to U:\;U⊲{\left\{\;TS,\;b\cdot P\;\right\}}_{K_{hmac}}$

#### 5.1.2. Logical Assumptions

We made the following assumptions according to the nature of the protocol. First, the CN believes that the UE should control the HMAC key issued by themselves:(1)$$C\;\left|\equiv \right.\;U\;\left|\Rightarrow \right.\;U\overset{K_{hmac}}{↔}C$$

Second, since both the CN and the UE check the timestamp in the protocol, they should believe that the timestamps are fresh:(2)$$C\left|\equiv \right.\#\left(TS\right)$$
(3)$$U\left|\equiv \right.\#\left(TS\right)$$

Third, the CN and the UE should also believe that their locally generated keys are trustworthy to themselves:(4)$$U\;\left|\equiv \right.\;U\overset{K_{hmac}}{↔}C$$
(5)$$U\;\left|\equiv \right.\;a$$
(6)$$U\;\left|\equiv \right.\;a\cdot P$$
(7)$$C\;\left|\equiv \right.\;b$$
(8)$$C\;\left|\equiv \right.\;b\cdot P$$

Fourth, the UE should believe that the key generated by the CN is controlled and trusted by himself. Similarly, the CN should also believe that the keys generated by UE are controlled by himself:(9)$$U\;\left|\equiv \right.\;C\;\left|\Rightarrow \right.\;b\cdot P$$
(10)$$C\;\left|\equiv \right.\;U\;\left|\Rightarrow \right.\;a\cdot P$$
(11)$$U\;\left|\equiv \right.\;C\;\left|\equiv \right.\;b$$
(12)$$C\;\left|\equiv \right.\;U\;\left|\equiv \right.\;a$$

Fifth, the CN should believe the secret of the one-time hash sent from the UE by validating it with the records in the blockchain. Additionally, since it is only valid once, it can be viewed as a fresh nonce:(13)$$C\;\left|\equiv \right.\;U\overset{Y}{⇋}C$$
(14)$$C\;\left|\equiv \right.\;\#\left(Y\right)$$

Finally, since ECDH is used, it can be assumed that for the UE (*U*), the session key $U\overset{K_{UC}}{↔}C=a\cdot b\cdot P$ can be calculated with the received b·P and the locally generated a. Similarly, for the CN (*C*), the session key $U\overset{K_{UC}}{↔}C=b\cdot a\cdot P$ can be calculated with the received a·P and the locally generated b.

#### 5.1.3. Protocol Goal

The goal of the 5GSBA is to achieve mutual authentication between two sides (UE and CN). Hence, we need to create a mutually trusted common session key after the execution of the protocol. We can express the goal with these four equations:(15)$$U\;\left|\equiv \right.\;U\overset{K_{UC}}{↔}C$$
(16)$$C\;\left|\equiv \right.\;U\overset{K_{UC}}{↔}C$$
(17)$$U\;\left|\equiv \right.\;C\;\left|\equiv \right.\;U\overset{K_{UC}}{↔}C$$
(18)$$C\;\left|\equiv \right.\;U\;\left|\equiv \right.\;U\overset{K_{UC}}{↔}C$$

#### 5.1.4. Protocol Verification

The detailed verification steps are listed below. Using the rule with Message 1, we have Equation (19):(19)$$\frac{C\;\left|\equiv \right.\overset{PK_{core}}{\mapsto }\;U,\;C⊲{\left\{\langle U\overset{K_{hmac}}{↔}C,\;{Y\rangle }_{Y}\right\}}_{PK_{core}}\;}{C⊲\langle U\overset{K_{hmac}}{↔}C,{Y\rangle }_{Y}}$$

Using the message meaning rule with Equations (13) and (19), we have Equation (20):(20)$$\frac{\;C\left|\equiv \right.\;U\overset{Y}{⇋}C,\;C⊲\langle U\overset{K_{hmac}}{↔}C,\;{Y\rangle }_{Y}}{C\left|\equiv \right.\;U\;|\sim \;\left(U\overset{K_{hmac}}{↔}C,\;Y\right)}$$

Using the freshness rule with Equations (14) and (19), we have Equation (21):(21)$$\frac{\;C\left|\equiv \right.\#\left(Y\right)}{C\;\left|\equiv \;\right.\#\left(U\overset{K_{hmac}}{↔}C,\;Y\right)}$$

Using the nonce verification rule with Equations (20) and (21), we have Equation (22):(22)$$\frac{\;C\left|\equiv \right.\#\left(U\overset{K_{hmac}}{↔}C,\;Y\right),\;C\left|\equiv \right.\;U\;|\sim \;\#(K_{hmac},\;Y)}{C\;\left|\equiv \;\right.U\;\left|\equiv \;\right.\left(U\overset{K_{hmac}}{↔}C,\;Y\right)}$$

Using the belief rule with Equation (22), we have Equation (23):(23)$$\frac{\;C\;\left|\equiv \right.\;U\;\left|\equiv \right.\;\left(U\overset{K_{hmac}}{↔}C,\;Y\right)}{C\;\left|\equiv \;\right.U\;\left|\equiv \;\right.U\overset{K_{hmac}}{↔}C}=C\;\left|\equiv \right.\;U\;\left|\equiv \right.\;U\overset{K_{hmac}}{↔}C$$

Using the jurisdiction rule with Equations (1) and (23), we have Equation (24):(24)$$\frac{C\;\left|\equiv \right.\;U\;\left|\Rightarrow \right.\;U\overset{K_{hmac}}{↔}C,\;\;\;C\;\left|\equiv \right.\;U\;\left|\equiv \right.\;U\overset{K_{hmac}}{↔}C}{C\;\left|\equiv \right.\;U\overset{K_{hmac}}{↔}C}=C\;\left|\equiv \right.\;U\overset{K_{hmac}}{↔}C$$

As a result, the CN believes the received HMAC key, so the CN continues to process Message 1. Using the message meaning rule with Equation (24) and Message 1, we have Equation (25):(25)$$\frac{C\;\left|\equiv \right.\;U\overset{K_{hmac}}{↔}C,\;C⊲{\left\{\;TS,\;a\cdot P\;\right\}}_{K_{hmac}}\;}{C\;\left|\equiv \right.\;U\;|\sim \;\left(TS,\;a\cdot P\right)}=C\;\left|\equiv \right.\;U\;|\sim \left(TS,\;a\cdot P\right)$$

Using the freshness rule with Equations (2) and (25), we have Equation (26):(26)$$\frac{C\;\left|\equiv \right.\;\#\left(TS\right)}{C\;\left|\equiv \right.\;\#\left(TS,\;a\cdot P\right)}=C\;\left|\equiv \right.\;\#\left(TS,\;a\cdot P\right)$$

Using the nonce verification rule with Equations (25) and (26), we have Equation (27):(27)$$\frac{C\;\left|\equiv \right.\;\#\left(TS,\;a\cdot P\right)\;,\;\;\;C\;\left|\equiv \right.\;U\;\left|\sim \right.\left(TS,\;a\cdot P\right)}{C\;\left|\equiv \right.\;U\;\left|\equiv \right.\;\left(TS,\;a\cdot P\right)}$$

Using the belief rule with Equation (27), we have Equation (28):(28)$$\frac{C\;\left|\equiv \right.\;U\;\left|\equiv \right.\;\left(TS,\;a\cdot P\right)}{C\;\left|\equiv \right.\;U\;\left|\equiv \right.\;a\cdot P}=C\;\left|\equiv \right.\;U\;\left|\equiv \right.\;a\cdot P$$

Using the jurisdiction rule with Equations (10) and (28), we have Equation (29):(29)$$\frac{C\;\left|\equiv \right.\;U\;\left|\Rightarrow \right.\;a\cdot P,\;C\;\left|\equiv \right.\;U\;\left|\equiv \right.\;a\cdot P}{C\;\left|\equiv \right.\;a\cdot P}=C\;\left|\equiv \right.\;a\cdot P$$

As a result, the CN believes the received ECDH public key from UE, and the protocol continues with Message 2. Using the message meaning rule with Equation (4) and Message 2, we have Equation (30):(30)$$\frac{U\;\left|\equiv \right.\;U\overset{K_{hmac}}{↔},\;P\;⊲{\left\{TS,\;\;b\cdot P\right\}}_{K_{hmac}}}{U\;\left|\equiv \right.\;C\;|\sim \;\left(TS,\;\;b\cdot P\right)}=U\;\left|\equiv \right.\;C\;|\sim \;\left(TS,\;b\cdot P\right)$$

Using the freshness rule with Equations (3) and (30), we have Equation (31):(31)$$\frac{U\;\left|\equiv \right.\;\#\left(TS\right)}{U\;\left|\equiv \right.\;\#\left(TS,\;\;b\cdot P\right)}=U\;\left|\equiv \right.\;\#\left(TS,\;b\cdot P\right)$$

Using the nonce verification rule with Equations (30) and (31), we have Equation (32):(32)$$\frac{U\;\left|\equiv \right.\;\#\left(TS,\;b\cdot P\right),\;U\;\left|\equiv \right.\;C\;|\sim \;\left(TS,\;b\cdot P\right)}{U\;\left|\equiv \right.\;C\;\left|\equiv \right.\;\left(TS,\;b\cdot P\right)}$$

Using the belief rule with Equation (32), we have Equation (33):(33)$$\frac{U\;\left|\equiv \right.\;C\;\left|\equiv \right.\;\left(TS,\;b\cdot P\right)}{U\;\left|\equiv \right.\;C\;\left|\equiv \right.\;b\cdot P}=U\;\left|\equiv \right.\;C\;\left|\equiv \right.\;b\cdot P$$

Using jurisdiction rule with Equations (9) and (33), we have Equation (34):(34)$$\frac{U\;\left|\equiv \right.\;C\;\left|\Rightarrow \right.\;b\cdot P,\;U\;\left|\equiv \right.\;C\;\left|\equiv \right.\;b\cdot P}{U\;\left|\equiv \right.\;b\cdot P}=U\;\left|\equiv \right.\;b\cdot P$$

As a result, the UE also believes the received ECDH public key from the gNB.

For the UE, since we know $U\;\left|\equiv \right.\;a$ in Equation (5) and $U\;\left|\equiv \right.\;b\cdot P$ in Equation (34), the common key $U\overset{K_{UC}}{↔}C$ can be derived in Equation (35):(35)$$U\;\left|\equiv \right.\;a\cdot \left(b\cdot P\right)=U\;\left|\equiv \right.\;a\cdot b\cdot P=U\;\left|\equiv \right.\;U\overset{K_{UC}}{↔}C$$

For the CN, since we know $C\;\left|\equiv \right.\;b$ in Equation (7) and $C\;\left|\equiv \right.\;a\cdot P$ in Equation (29), the common key $U\overset{K_{UC}}{↔}C$ can be derived in Equation (36):(36)$$U\;\left|\equiv \right.\;a\cdot \left(b\cdot P\right)=U\;\left|\equiv \right.\;a\cdot b\cdot P=U\;\left|\equiv \right.\;U\overset{K_{UC}}{↔}C$$

Moreover, to finish the protocol, the CN has to believe Message 1 to continue the protocol and send Message 2. Hence, we can say that if the UE has received Message 2, the UE can be sure that the CN has believed Message 1, and therefore $U\;\left|\equiv \right.\;C\;\left|\equiv \right.\;a\cdot P$. Combining this with Equation (11), we can conclude that:(37)$$U\;\left|\equiv \right.\;C\;\left|\equiv \right.\;b\cdot \left(a\cdot P\right)=U\;\left|\equiv \right.\;C\;\left|\equiv \right.\;a\cdot b\cdot P=U\;\left|\equiv \right.\;C\;\left|\equiv \right.\;U\overset{K_{UC}}{↔}C$$

Similarly, the UE has to believe Message 2 to start the subsequent data transmission. Hence, we can say that if the CN receives the subsequent data correctly, CN can be sure that the UE has believed Message 2, and therefore $C\;\left|\equiv \right.\;U\;\left|\equiv \right.\;b\cdot P$. Combining this with Equation (12), we can conclude that:(38)$$C\;\left|\equiv \right.\;U\;\left|\equiv \right.\;a\cdot \left(b\cdot P\right)=C\;\left|\equiv \right.\;U\;\left|\equiv \right.\;a\cdot b\cdot P=C\;\left|\equiv \right.\;U\;\left|\equiv \right.\;U\overset{K_{UC}}{↔}C$$

Consequently, all the security goals are satisfied. Hence, the security of 5GSBA is logically verified.

### 5.2. Scyther Tool

The Scyther tool [33] is an automated formal verification tool for analyzing authentication protocols. Under the perfect cryptography assumption and the Dolev–Yao adversary model [30], Scyther searches for all potential security vulnerabilities of a protocol efficiently. Perfect cryptography assumption refers that the cryptographic functions used in the protocol are assumed to be secure. Adversaries should know nothing about the encrypted content unless they hold the decryption key. The Dolev–Yao adversary model, as mentioned in Section 3.2, assumes that there are neighboring attackers in the network. In this section, we model the 5GSBA with the Security Protocol Description Language (SPDL) and let Scyther find all security issues automatically.

The Scyther tool has six different security claims: **Aliveness** or **Alive** ensure the protocol instances complete their steps with any active responders. That is, all replies should be active replies from living partners, not replayed messages. **Niagree** ensures a protocol instance receives the expected variables without consideration of a one-to-one relationship (i.e., non-injective agreement). **Nisynch** ensures the protocol can complete a run as expected without a one-to-one relationship (i.e., non-injective synchronization). **Weakagree** ensures all protocol instances communicate with their same set of initiators or responders (i.e., injective). **Reachable** is the checkpoint claim indicating that the protocol can reach the specific line, which can be used for code debugging. **Secrecy** or **SKR** check if the specified variables or session keys remain secret to adversaries throughout the execution of the protocol. By combining all these security claims, Scyther ensures the injective agreement of the protocol and detects most of the network protocol attacks including message fabrication, message replay, and MITM attacks.

Our formal verification result with Scyther is shown in Figure 4. Specifically, we created a SPDL model that has two roles: the gNB and the UE. First, the 5GSBA can achieve a mutual key agreement using ECDH. To emulate ECDH key exchange between two parties, we define functions g1 and g2, and then set alpha to be g1(a) and beta to be g1(b). We firstly use the claims of Secret a and Secret b to check the secrecy of the ECDH private keys. Then, we emulate the derivation of the ECDH common key with g2(beta, a) (i.e., a · b · G) and g2(alpha, b) (i.e., b · a · G). We also verify the secrecy of this ECDH common key with *SKR* claims. Second, the 5GSBA uses HMAC to ensure the integrity of all messages. To make sure adversaries cannot obtain the HMAC key $K_{hmac}$ (i.e., only gNBs can receive the key from the UE), we check it with Secret Khmac claims for both the gNB and the UE. Third, the 5GSBA can guarantee device anonymity by encrypting the SUPI into SUCI. To ensure the adversaries have no way to retrieve the SUPI of the UE through the air, we check the secrecy of the SUPI using the Secret MSIN claim. In this scenario, Mobile Subscriber Identification Number (MSIN) is equivalent to SUPI, because SUPI should contain mobile country code (MCC), mobile network code (MNC), and MSIN. While MCC and MNC are not the unique identifiers of a device, we only mandate MSIN to be secret. Fourth, our protocol assumes that the UE needs to use its one-time hash secret Y to prove its identity to gNB. To ensure that the adversaries cannot obtain the secret Y with any means, we checked its secrecy using Secret Y claims. Additionally, the secrecy of Y2 was checked to ensure that the UE could not reveal the next one-time hash secret by any means. Fifth, to further ensure the correctness of our SPDL model, a *Reachable* claim was put at the end of every role. This made sure that every line of our code had been executed. Finally, by testing all security claims including *Aliveness*, *Niagree*, *Nisynch,* and *Weakagree* in both parties, this showed that the 5GSBA guarantees injective agreement with active initiator and responders. In other words, we conclude that no network attacks were found in the 5GSBA.

### 5.3. Security Analysis

We qualitatively justify that our 5GSBA provides the following security features.

**Mutual Authentication and Key Agreement:** By the 5GSBA, UE uses a one-way hash secret Y stored in the USIM to prove that it is the legitimate party. A gNB verifies it by comparing it to the one-way hash value $H\left(Y\right)$ stored in the private blockchain. Since finding the original secret of a hash is a computationally infeasible problem, the UE can effectively prove to the gNB that its authentication request message is legitimate. Besides, the UE encrypts the one-way hash secret Y and a randomly generated HMAC key $K_{hmac}$ using the public key $PK_{core}$. Since only legitimate gNBs possessing the private key $SK_{core}$ can read the one-way hash secret and the HMAC key, by issuing a correct HMAC code σ2, the gNB effectively prove to the UE that the authentication response is also legitimate. Thus, both gNB and UE can be mutually authenticated to derive a common key using ECDH parameters in the authentication messages.

**Secure Data Transmission**: The 5GSBA assumes that all subsequent user data will be encrypted with the session key generated in the authentication procedure. For the session key generation, a UE sends an ECDH public key a⋅P to a gNB, and the gNB replies to another ECDH public key b⋅P to the UE. Hence, the actual session key a⋅b⋅P is never transmitted anywhere. In fact, the ECDH relies on the Computational Diffie–Hellman (CDH) problem, which means it is difficult to find the a from a⋅P. Even if an adversary captures all the authentication messages, it is computationally infeasible for him to recover the session key by finding either a or b to calculate the a⋅b⋅P. Consequently, the 5GSBA can ensure only legitimate parties (i.e., UE and gNB) can read the user data.

**Session Key Perfect Forward/Backward Secrecy:** The 5GSBA uses the public key $PK_{core}$ of the 5GC, the randomly generated HMAC key $K_{hmac}$, and the one-way hash secret Y for authentication purposes only. The session key generation relies on the randomly generated ECDH parameters. Hence, when the permanent keys are stolen in the future, attackers still could not derive the session key to recover the content of a specific session. Additionally, since the newly generated ECDH parameters for each session are irrelevant to the previous or future sessions, compromising the current session key will only affect the current session. The secrecy of the previous or future sessions will remain unaffected.

**Device Anonymity:** Since the SUCI is mandatory by the 5GSBA, the SUPIs of requesting UE devices are always concealed with ECIES. Hence, eavesdroppers cannot use authentication request messages to identify or trace any devices.

**Protocol Attack Resistance:** The 5GSBA outperforms most proposals, including the standardized 5G-AKA with the stronger resistibility to many attacks. For example, thanks to the aforementioned security properties, the 5GSBA prevents common attacks such as eavesdropping, location tracking, and man-in-the-middle (MITM) attacks. Moreover, it is highlighted that some critical attacks can be prevented including DoS/DDoS attacks, linkability attacks, UE impersonation attacks, rogue base station attacks, and replay attacks:**DoS Semantic Attack****Prevention:** With the 5GSBA, semantic attacks exploiting the weaknesses of the protocol are impracticable, because the authentication request contains a timestamp and a one-time hash secret Y. Specifically, when the gNB receives an authentication request, it first checks if the timestamp is fresh. Then, it decrypts the SUCI and compares the Y with the records stored in the private blockchain. If the calculated hash value does not match, it will reject the session immediately. In this way, adversaries cannot hoard multiple sessions in the gNB by replaying the same authentication messages (i.e., the original SUCI in 5G-AKA). Additionally, since the authentication request involves only the computationally inexpensive ECIES and hash functions, adversaries cannot exhaust the computational resources of gNBs easily;**DDoS Flooding Attack****Prevention:** By decentralizing the authentication tasks from the AUSF/UDM to all gNBs, the 5GSBA lessens the effects of flooding attacks that paralyze the network with authentic requests. In the 5G era, the inter-site distance (ISD) of gNBs is getting smaller, and the number of gNBs deployed keeps growing. Hence, the total computational power of gNBs is growing steadily, making it increasingly difficult to flood or even paralyze the entire 5G network. Furthermore, since the 5GSBA shifts the authentication tasks from AUSF/UDM to gNBs, it can also prevent the single-point-of-failure. One gNB failure or one AUSF failure will not affect the entire 5G network. Thus, DDoS attacks in the 5G authentication can be prevented, and the quality of service (QoS) across the 5G network can be maintained. The performance analysis will show that the 5GSBA can serve much more incoming authentication requests than the existing centralized schemes;**Linkability Attack Prevention:** Unlike conventional symmetric key-based protocols such as 5G-AKA, the 5GSBA generates session keys using ECDH instead of the sequence number. Hence, the 5GSBA does not have the MAC failure or synchronization failure commonly found in symmetric key-based AKA protocols. Adversaries can no longer use these error messages as a loophole to trace a specific device;**UE Impersonation Attack Prevention:** By the conventional 5G-AKA protocol, users have to trust the network operator implicitly. Since the network operator owns a copy of users’ symmetric keys and sequence numbers, insider attackers in the network can impersonate the UE by abusing these keys. By the 5GSBA, since the one-time hash secret is only stored at the USIM, there is no way for network operators to impersonate the UE using the data stored in the private blockchain. Hence, if the network operators cannot provide the one-time hash secret used in the authentication, users can simply deny all malicious behavior for that session;**Rogue Base Station Attack Prevention:** By the 5GSBA, since only the legitimate gNBs can decrypt the SUCI, rogue base stations cannot produce genuine authentication responses by generating the correct HMAC code σ. Thus, it can prevent UE from establishing connections with rogue base stations;**Replay Attack Prevention:** The authentication requests and responses by the 5GSBA are all tagged with a timestamp TS, and the HMAC key $K_{hmac}$ should also work once only. Therefore, by checking the timestamp in both UEs and gNBs, replayed messages can be easily identified and discarded;

**Battery Depletion Attack Prevention:** With the 5G-AKA, UE must respond to identity requests from the serving network by generating a fresh SUCI. If some rogue base stations frequently send the identity requests, the batteries of UE devices could deplete faster. In the 5GSBA, only UE can take the initiative to generate a fresh SUCI for authentication. Hence, the serving network cannot force the UE to create a fresh SUCI, and it prevents the battery depletion attacks effectively.

## 6. Performance Evaluation

Most UE has limited computational resources and battery life. Additionally, the network resources in the 5G network are always finite. Hence, a good authentication scheme should have low computational overhead, communication overhead, and energy consumption. In this section, we analyze the performance of the 5GSBA from these three aspects. Additionally, we evaluate its effectiveness against DoS/DDoS attacks using two simulation experiments. For the comparison, 5G-AKA [4] is selected because it is the 3GPP standardized protocol. SE-AKA [20] is chosen because it supports perfect forward secrecy with the similar security functions as the 5GSBA. Another blockchain-based 5G authentication protocol, BB-AKA 5G [10] is also included to show that the 5GSBA can achieve a better performance than the existing blockchain-based scheme.

### 6.1. Computational Overheads

#### 6.1.1. Theoretical and Experimental Delays

A smaller computational delay means the protocol can run faster, so it is always preferred. In this subsection, computation overheads are evaluated through simulation experiments. All simulations are conducted on a computer with Intel^®^ Core™ i5-3210M CPU @ 2.5 GHz CPU and 16 GB of RAMs. The Charm Crypto v0.5 with Python 3.7.5 is used to create and run the simulation. Charm Crypto [34] is a Python wrapper of Pairing-Based Cryptography (PBC) libraries. It allows the easy prototyping of cryptographic functions based on elliptic curve cryptosystems. Among all the selected protocols, each of them uses various cryptographic functions, and each of them has a unique key size requirement. To holistically and equally evaluate their performance at the same security level, we conformed to the recommendation from NIST [35] to use 256-bit equivalent key strength throughout the simulations. Hence, for all elliptic curve cryptography (ECC)-based operations, secp256k1 was chosen as the default elliptic curve. For HMAC operations, HMACSHA256 was selected. We ran each cryptographic function 2000 times to measure its average time. Moreover, to evaluate the data access delays incurred in blockchains, we followed the specifications in Section 4.2 to create a blockchain prototype. This blockchain had many 1-megabyte-sized blocks. Every block was stored as one file, and all transactions in the blockchain were indexed using Python Dictionary. To compare the performance difference between blockchain and traditional database, we also built another centralized database with MariaDB v10.4.14. Finally, Table 4 shows the experimental time of execution for the different functions. Then, all this information is combined with the theoretical computational times in Table 5 to find the experimental computational overheads shown in Figure 5.

It is noteworthy that, for all blockchain-based schemes, we omitted the blockchain indexing time in the computational overhead calculation. Blockchain indexing is a process for gNBs to categorize recently downloaded blocks. It can improve the blockchain random access speed to a constant time. Since indexing can be performed in the background parallelly at any time, it does not impose a significant negative effect on authentication. Hence, only the average blockchain transaction read time $T_{BC.read}$ and write time $T_{BC.write}$ were included in the calculation.

For the experimental computational delays shown in Figure 5, the 5GSBA was slower than the standardized 5G-AKA by 5.1377 − 3.4683 = 1.6694 milliseconds. This is because the 5GSBA employs asymmetric ECDH to replace the less secure symmetric key-based key derivation function (KDF) in the 5G-AKA [4]. By introducing this tiny computational overhead, the 5GSBA can ensure session key perfect forward secrecy (PFS), backward secrecy, and linkability attack prevention. On the other hand, the computational time of the 5GSBA and the SE-AKA [20] are comparable because both schemes employ ECDH, but the 5GSBA offers an enhanced security function in terms of DoS attack prevention and DDoS alleviation. The BB-AKA 5G [10] consumes much more time than the 5GSBA because the BB-AKA 5G needs two ECDSA signatures and verifications at the gNB and the AMF, while in contrast, the 5GSBA only requires one lightweight one-way hash secret verification in the gNB. Thus, the 5GSBA saves (10.4107 − 5.1377)/10.4107 = 50.6% of computational overhead compared to the BB-AKA 5G. Consequently, although the 5GSBA protocol has unavoidably introduced tiny computational overhead, it is still the most efficient protocol in terms of balancing between security and performance.

#### 6.1.2. Average Delays under Unknown Attacks

Although the 5GSBA could resist several malicious attacks, as shown in the security analysis, it is possible that new unknown attacks in the 5G network could interrupt the authentication process. To evaluate the performance under unknown attacks, it was assumed that, for each step of the protocol, the network faced either known or unknown attacks. Specifically, there will be a probability that the protocol will encounter an unknown attack. If the incoming attack is known, the protocol should continue smoothly until completion. However, if the incoming attack is unknown, the protocol would be unavoidably interrupted. In this case, it must restart from the first step until it completes the last step. Based on the assumptions above, we created a simulation model on MATLAB 2020b. Using 1 million threads to run the protocol, we found the average execution time to complete the protocol under different unknown attacks, as shown in Figure 6. The results showed that although the 5GSBA had slightly higher delays (from 1.67 to 4.17 milliseconds) than those of the 5G-AKA [4] in all scenarios, its performance was still much better than another blockchain-based protocol of the BB-AKA 5G [10]. Considering that the 5GSBA ensured the best security among all protocols, we can still conclude that it can achieve a reasonable balance between high security and reasonable delays.

### 6.2. Communication Overhead

As bandwidth resources in the 5G network are invaluable, smaller communication overhead is always preferred. Thus, we further evaluated the communication overhead in terms of bandwidth consumption and transmission overhead.

For the bandwidth consumption based on five different security levels suggested by NIST [35], the total message sizes of the related protocols are derived in Figure 7. It shows that our 5GSBA outperformed all other schemes by achieving the lowest bandwidth consumption. This is because the 5GSBA authenticates the incoming UE locally using the two-step protocol, which can save redundant forwarding messages.

For communication delays there are two components: propagation delays and transmission delays. Propagation delay calculations consist of both wired and wireless connections. For the wired connections, we referred to the link between the gNB and the nearest 5GC functional entities. It is assumed that optical fiber is deployed for all wired connections, and the 5GC functional entities are geographically distant to the gNB. To simulate a real-life situation, we assumed the distance between the gNB and the nearest 5GC functional entity to be 1 km, and the distance between two functional entities within the 5GC to be less than 0.5 km. For the wireless connection, according to the dense urban 5G service requirement in TS 22.261 [36], we assumed that the inter-site distance (ISD) between two gNBs was 200 m, and the requesting UE was located to the edge of the coverage of the gNB. Therefore, the distance between the gNB and UE was about 100 m. The transmission delay calculation also consisted of wired and wireless connections. For the wired connections, due to the link of optical fiber, it was assumed that the uplink and downlink speed were both 1 Gbit/s. For the wireless connections, similarly, by following TS 22.261 [36], it was assumed that the uplink speed for UE was 50 Mbit/s and the downlink speed for UE was 300 Mbit/s. For the blockchain-based schemes, it was assumed that blockchain nodes could synchronize new blocks parallelly during the idle time of gNBs. To better reflect the extra communication overhead introduced by blockchains, we included the average delay of downloading one old transaction and uploading one new transaction into the calculation. Finally, by combining all the assumptions above, the sums of propagation and transmission delays are derived in Figure 8.

Figure 8 shows that the 5GSBA achieves similar total communication delays from 160-bit to 256-bit key length. Although the total communication delay of the 5GSBA was marginally higher than that of the 5G-AKA [4] from 384 bit to 512 bit, the 5GSBA still consumed much less time than that of the BB-AKA 5G [10]. This is because the 5GSBA saved some propagation delays by consuming the lengthy authentication requests locally in gNBs. Considering that the 256-bit ECC key was accepted in the NIST recommendation [35], the communication delays of the 5GSBA were very close to that of the 5G-AKA. Overall, the 5GSBA achieved the right balance between strong security and relatively low communication overhead.

### 6.3. Energy Consumption for UE

Since mobile devices have limited battery life, a security protocol with the strongest security and the least energy consumption is always preferable. To evaluate the energy consumption for UE two factors should be considered: data transmission energy and the energy for the execution of the cryptographic functions. For the data transmission energy, the data transfer power model in [37] is adopted to estimate the energy consumption. Thus, the energy cost for uplink transmission of the UE is in Equation (39) and the energy cost for the downlink is in Equation (40), where $\alpha_{u}$= 438.39 mW/Mbps, $\alpha_{d}$ = 51.97 mW/Mbps, β = 1288.04 mW, $t_{udr}$ is the uplink throughput, $t_{ddr}$ is the downlink throughput, $t_{ul}$ is the transmission time spent on the uplink, and $t_{dl}$ is the transmission time spent on the downlink. We further assume that the uplink and downlink throughput follow the dense urban 5G service requirement in TS 22.261 [36], so that $t_{udr}$ = 50 Mbps and $t_{ddr}$ = 300 Mbps.
(39)$$E_{ul}=\left(\alpha_{u}t_{udr}+\beta \right)\cdot t_{ul}\;$$
(40)$$E_{dl}=\left(\alpha_{d}t_{ddr}+\beta \right)\cdot t_{dl}\;$$

On the other hand, for the energy consumption for the execution of cryptographic functions, the measure of energy cost approximation in [38] is taken. In [38], all experiments were conducted using a battery-powered Compaq iPAQ H3670 PDA. It was equipped with an Intel SA-1110 StrongARM processor clocked at 206 MHz, 64 MB of RAM, and 16 MB of FlashROM. Moreover, the energy cost of ECDH public key generation $E_{ECDH.gen}$ was 276.7 mJ, the ECDH common key derivation $E_{ECDH.ck}$ was 163.5 mJ, the ECDSA signature generation $E_{ECDSA.sign}$ was 134.2 mJ, and the ECDSA signature validation $E_{ECDSA.ver}$ was 196.23 mJ. For the energy cost of symmetric key-based operations, the symmetric key-based encryption $E_{sym.}$ was 9.92+2.29l uJ, where l was the number of bytes of the cleartext. The HMAC operation $E_{HMAC}\;$was 1.16 mJ, and hash operation $E_{hash}$ was 0.76 mJ. For the ECIES operations, since it was a hybrid encryption combining both CDH problems and a key encapsulation mechanism (KEM), the cost of the ECIES encryption was estimated as $E_{ECDH.gen}+E_{ECDH.ck}+E_{sym}=440.20992+0.00229l$ mJ.

Finally, by combining all parameters above with the unknown attack model stated in Section 6.1.2, the average UE energy consumption for different protocols is simulated in Figure 9. It shows that all the schemes providing perfect forward and backward secrecy, including the 5GSBA [4], SE-AKA [20], and BB-AKA 5G [10], consume a similar amount of energy in a situation without any unknown attacks. When the unknown attack probability increases, the 5GSBA unavoidably uses more energy because of the increased SUCI generation. However, the 5GSBA still consumes less energy than the BB-AKA 5G. On the other hand, although the 5G-AKA uses the least energy, its power-saving symmetric key-based cryptosystem makes it vulnerable to many other known attacks.

### 6.4. Resistivity to DoS and DDoS Attacks

To show that our 5GSBA can fight against DoS and DDoS attacks, we further design two simulations to compare all related protocols. All simulation results show that the 5GSBA can achieve a high level of security functionality with reasonable delays.

#### 6.4.1. Average Delays under DDoS Flooding Attacks

One of the outstanding features of the 5GSBA is the alleviation of the flooding of DDoS attacks. The 5GSBA, since all authentication tasks are distributed at various gNBs, avoids congestion and processing at the single authentication entity AUSF. Hence, to illustrate the performance of the protocols under the flooding of many authentication requests, we can model all centralized protocols, such as 5G-AKA [4] and SE-AKA [20], as M/M/1 queuing models, while the blockchain-based protocol of the 5GSBA can be modeled as a M/M/c queuing model (c is the number of gNBs). For simplicity, it is assumed that the arrival and location of UE devices follow Poisson distribution, such that all devices within 1 km^2^ are located uniformly and can initiate authentication requests randomly. For the centralized protocols, since the AUSF in the 5GC should be a computationally powerful server, it is assumed that they run on the server 80% faster than the performance listed in Table 2. For blockchain-based protocols, as the computational resources for each gNB are limited, it is assumed that its computational time is the same as that in Table 2. Furthermore, one AUSF only serves the gNBs within the 1km^2^ area, and the ISD of every gNB is 200 m according to the TS 22.261 5G dense urban service requirement [36]. Consequently, each server needs to process the requests from $1\;{\mathrm{km}}^{2}/\pi 0.1^{2}=31.8\approx 32$ gNBs (i.e., c = 32).

Finally, the average time to complete the execution of the protocol is shown in Figure 10. It shows that although the decentralized 5GSBA takes slightly longer time than the other centralized schemes when the network traffic is low, as the number of requests increases, the 5GSBA outperforms the others by maintaining an almost constant delay. In other words, the 5GSBA can serve more authentication requests with the same amount of delay than the centralized schemes. In future, as the number of gNBs keeps increasing, it is expected that the 5GSBA will be able to perform even better by accommodating more users. Therefore, this shows that the 5GSBA is more robust in terms of serving more users, and thus is more effective in lessening the effects of flooding attacks. Moreover, since the 5GSBA can accommodate more requests, this also gives network operators more time to detect DDoS attacks and deploy countermeasures. For example, network operators can use the time to analyze network traffic and block potential DDoS attackers without impacting overall service quality. By doing so, attackers would find it more challenging to perform a successful DDoS attack in the 5G network.

#### 6.4.2. Average Successful Authentications under DoS Attacks

The 5GSBA can alleviate the effect of DoS attacks by verifying the timestamp and the HMAC code in the authentication request. To prove that the 5GSBA can provide the best DoS attack resistivity among all related schemes, a simulation model was built in MATLAB 2020b with the following assumptions. There are 1000 mobile devices sending authentication requests to a server at every one millisecond, which can hold—at most—1000 sessions simultaneously without any performance degradation. Then, for each authentication request, there could be a percentage chance that it is a fabricated DoS request. In case the request is legitimate, the server will complete the protocol operation normally following the protocol specifications. However, if the request is a DoS attack, either of the following results could happen. If the server can identify it as a DoS request, it will terminate the session immediately to accept another new request. If the server cannot identify it, the session will hold until it reaches the protocol expiry time at 3 s, which is a common default value for RADIUS server timeout. This simulation is run for 60 s. The number of successful authentications of different schemes is recorded in Figure 11.

Figure 11 shows that when there is no DoS attack in the network, the 5G-AKA performs the best among all schemes with the highest number of successful authentications. However, when the percentage of DoS attacks increases, the performance of the 5G-AKA [4] drops drastically, while the 5GSBA maintains the highest number of successful authentications. This is because the 5G-AKA has no way to identify if the authentication requests are fabricated, so the server wastes some sessions for holding until their expiry. On the other hand, the SE-AKA [20] can also provide DoS attack prevention, similar to our 5GSBA. However, the 5GSBA runs faster, so it has a higher number of successful authentications.

### 6.5. Discussion of the Results

The performance evaluation shows that when there are DoS or DDoS attacks in the 5G networks, the 5GSBA is the better protocol with a higher performance in terms of a stable authentication delay and a high authentication success rate. The only limitation of the 5GSBA would be the slightly added overhead in computational overhead, communication overhead, and energy consumption compared with the 3GPP 5G-AKA [4]. However, given that there will be an exponential growth of 5G mobile devices in the future, the risk of DoS and DDoS attacks will become more prominent. Thus, we believe that these imperceptible overheads are justifiable to safeguard future 5G networks.

## 7. Conclusions

In 5G networks, DoS and DDoS attacks have become a critical issue due to the increasing number of mobile devices. To ensure the robustness and security of the 5G network, we have proposed a Secure Blockchain-based 5G Authentication and Key Agreement (5GSBA) protocol in this paper. The 5GSBA protocol holds many security features that the existing 3GPP 5G-AKA scheme fails to achieve, including perfect forward secrecy, device anonymity, and most importantly, the resistivity to DoS and DDoS attacks. By the decentralization of authentication functions with blockchain technology, the 5GSBA delivers the best quality of service (QoS) among all schemes in relation to DoS and DDoS attacks. Our performance evaluation also demonstrates that the overhead added by 5GSBA is imperceptible compared to other existing solutions. Therefore, we believe that the 5GSBA protocol is ideal for balancing strong security functionality and high performance.

## Figures and Tables

**Figure 1 sensors-22-04525-f001:**
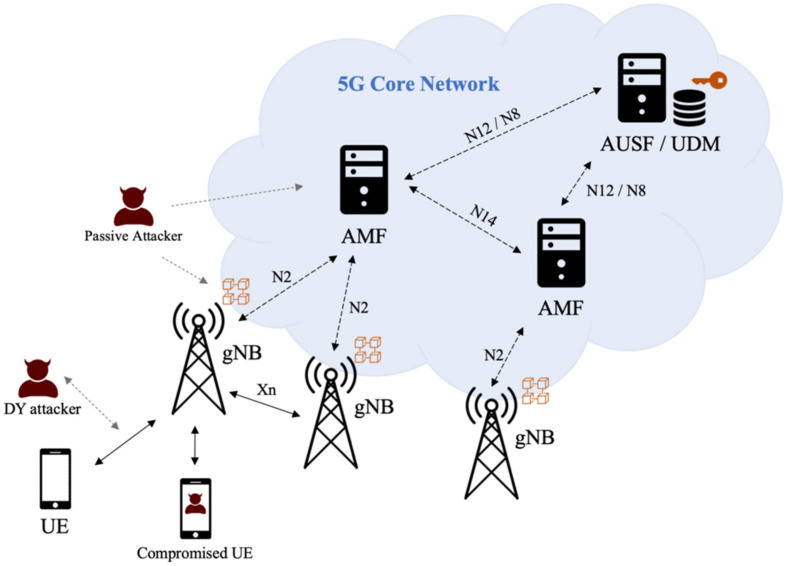
System and security model.

**Figure 2 sensors-22-04525-f002:**
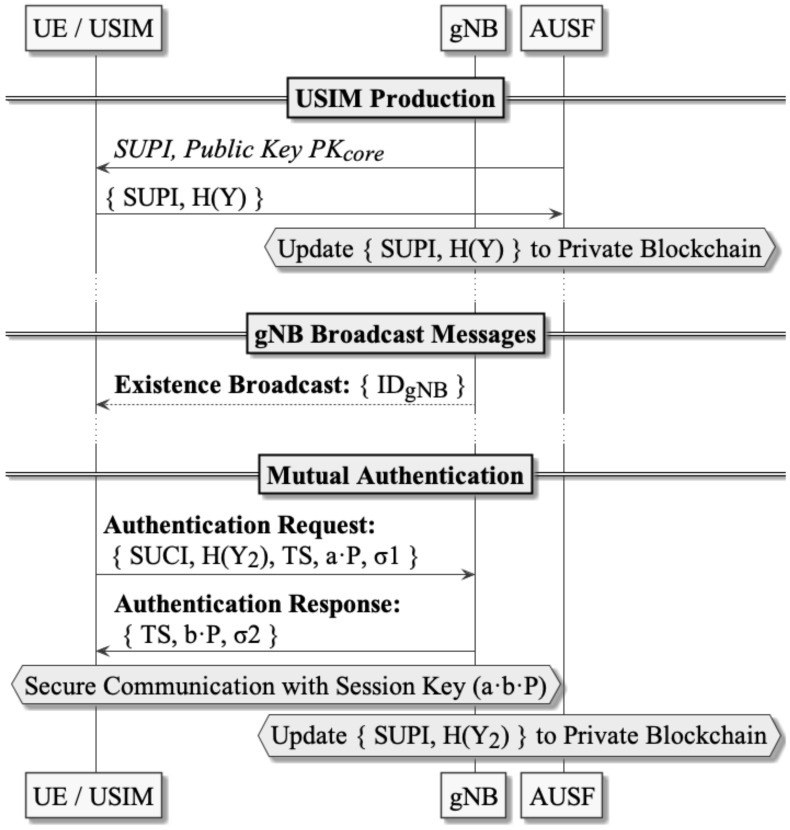
Sequence Diagram of 5GSBA.

**Figure 3 sensors-22-04525-f003:**
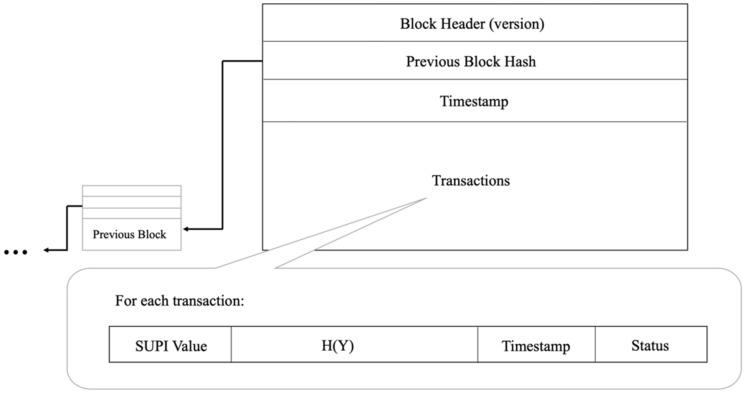
Structure of a block and a transaction.

**Figure 4 sensors-22-04525-f004:**
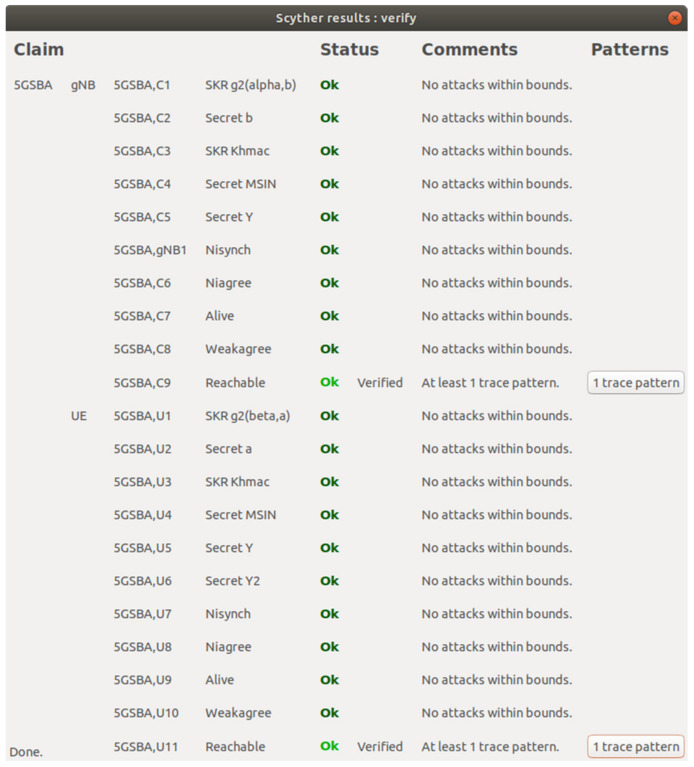
Formal verification with Scyther.

**Figure 5 sensors-22-04525-f005:**
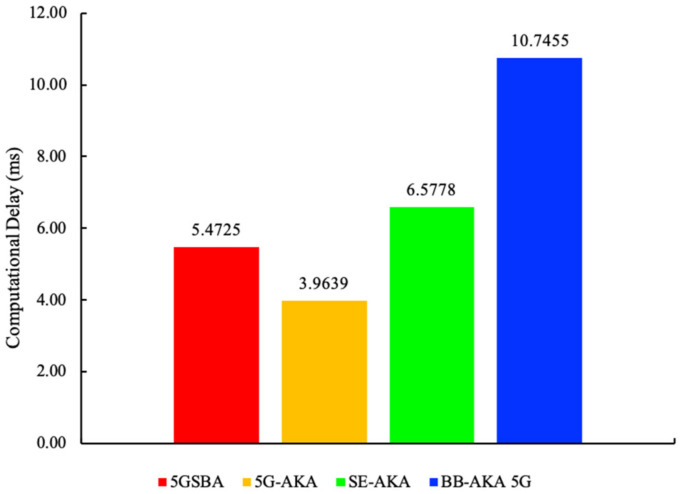
Experimental total computational delays.

**Figure 6 sensors-22-04525-f006:**
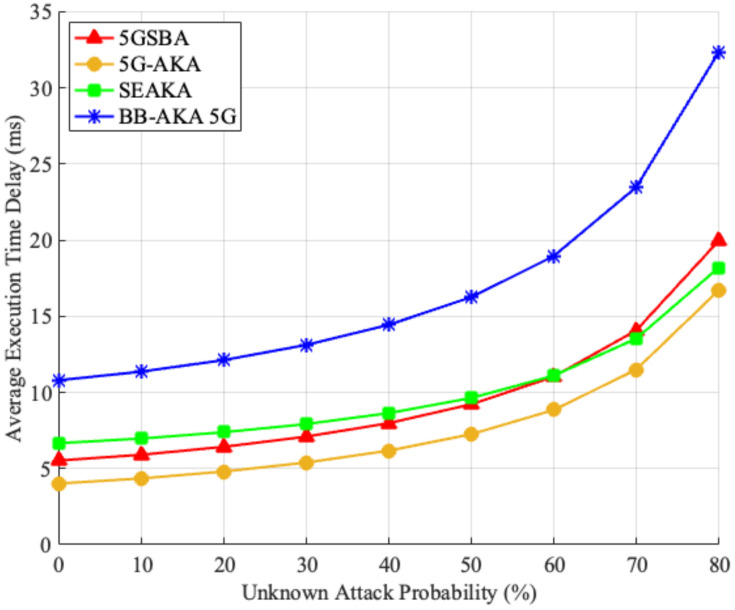
Average delays under unknown attacks.

**Figure 7 sensors-22-04525-f007:**
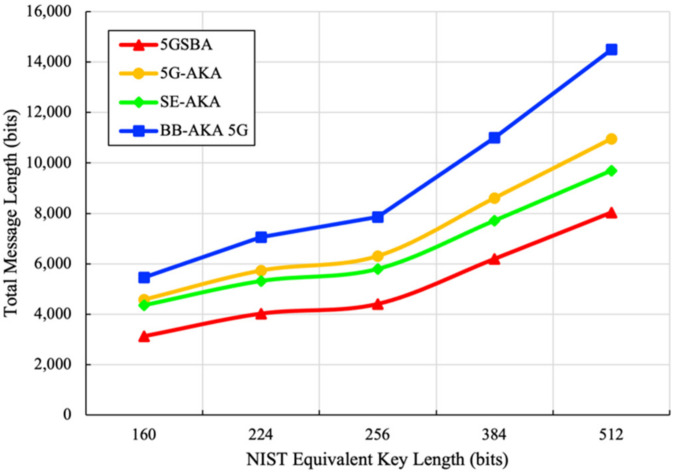
Total bandwidth consumption.

**Figure 8 sensors-22-04525-f008:**
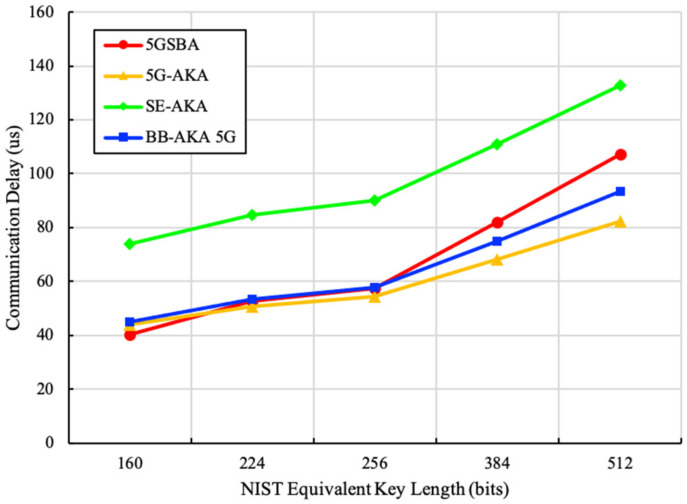
Total communication delays.

**Figure 9 sensors-22-04525-f009:**
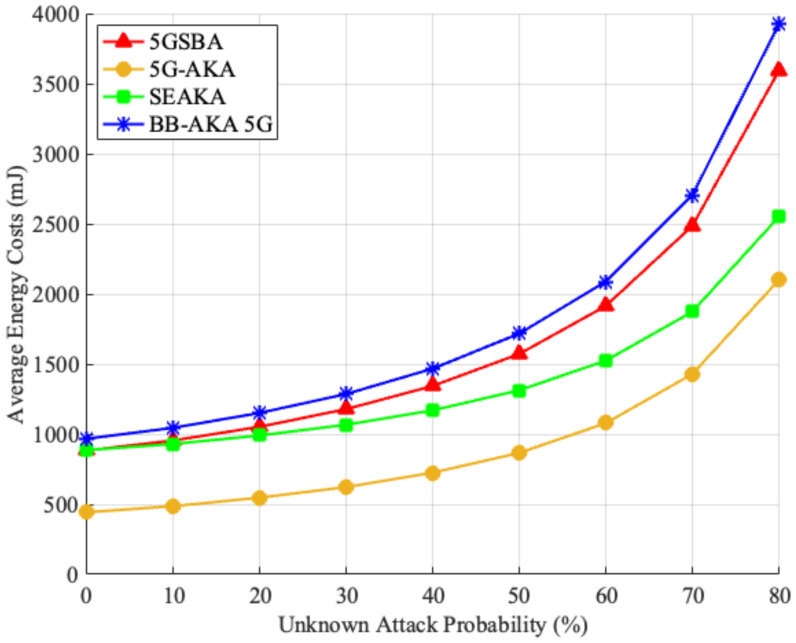
Energy consumption for UE.

**Figure 10 sensors-22-04525-f010:**
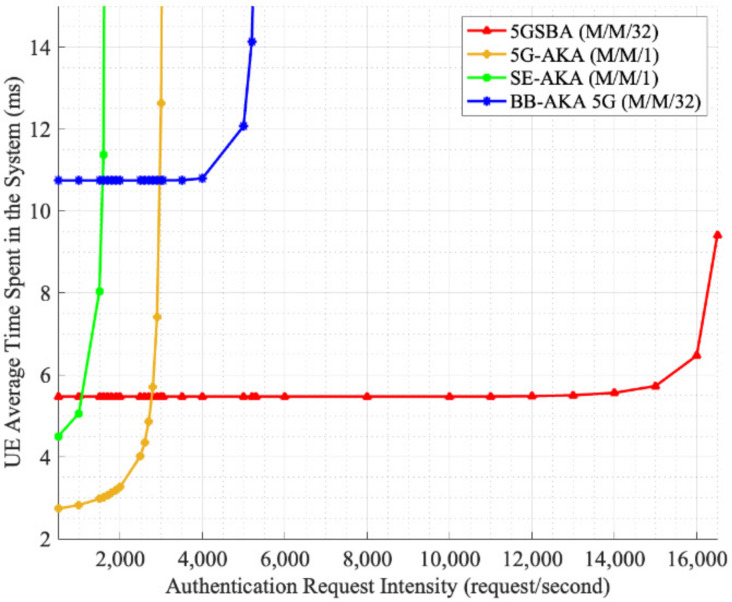
Average delay under many attacks.

**Figure 11 sensors-22-04525-f011:**
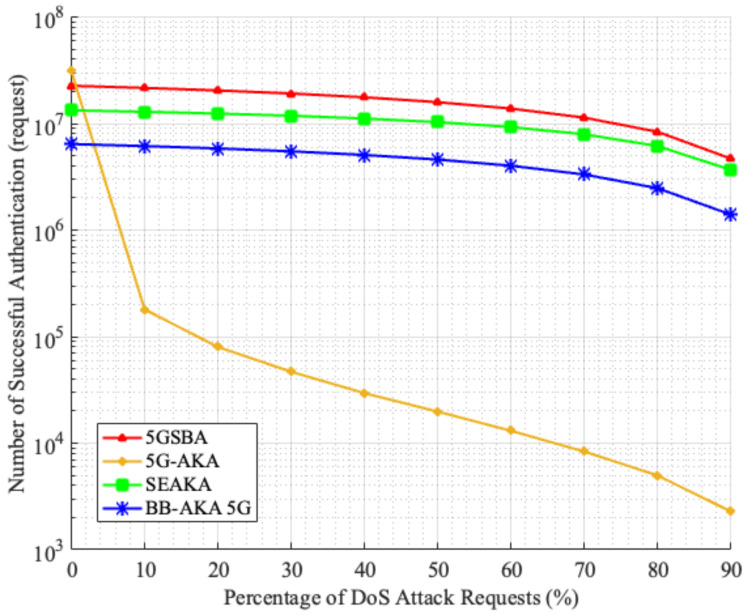
Successful authentications under DoS attacks.

**Table 1 sensors-22-04525-t001:** Recent works on blockchain-based 5G authentication protocols.

	Type	Highlights	Security Features					
			FV	DA	PFS	LA	DoS	DDoS
Our Work	5G Initial Authentication	Decentralized authentications with low overhead	✓	✓	✓	✓	✓	✓
[9]		No single trust authority					✓	✓
[10,11]		Decentralized authentication to nearby gNBs		✓	✓		✓	✓
[12]		Removal of obsolete data in blockchains		✓	✓		✓	✓
[13]		Inter-domain authentication					✓	✓
[14]		Improving 5G EAP-AKA’ protocol security				✓	✓	✓
[19]		Formally verified protocol using chameleon signature	✓	✓	✓		✓	✓
[15]	5G Handover Authentication	Efficient handover authentication	✓	✓	✓	✓	✓	
[16]		Optimized for frequent handover				✓		
[17]		Lightweight handover authentication				✓		
[18]		Traceability for base stations to record malicious devices	✓	✓	✓	✓		

FV = formally verified protocol; DA = device anonymity; PFS = perfect forward secrecy; LA = lightweight authentication; DoS = DoS attack prevention; DDoS = DDoS attack prevention.

**Table 2 sensors-22-04525-t002:** Recent works of AKA schemes with DoS and DDoS prevention.

	Method	Advantages	Drawbacks
[20]	Group Authentication	Prevention of DoS using timestamp	Weak DDoS prevention for individual authentication
[21]		Efficient group-based authentication	
[22]	Computation Pool	Fault-tolerant 5G-AKA authentication	5G-AKA is inherently vulnerable to DoS attacks
[23]	One-to-One Authentication	Lightweight and formally verified handover authentication	Vulnerable to DDoS due to centralized design
[24]		Formally verified protocol	
[25]		Lightweight symmetric key-based protocol	
[26]		Formally verified protocol	
[27]		Backward compatibility with 5G-AKA	
[28]		DDoS prevention using zero-knowledge proof	Centralized design, lack of formally verified protocol

**Table 3 sensors-22-04525-t003:** Notations.

Notation	Description
SUPI	Subscription Permanent Identifier of the UE
ID_gNB_	Permanent Identifier of gNB
P	Generator of the Elliptic Curve
Y/Y_2_	One-Time Hash Secret
H (msg)	Cryptographic Hash Function
HMAC (msg, K)	Keyed-hash Message Authentication Code
σ	Generated HMAC code
PK_core_	Public Key of 5G Core
SK_core_	Private Key of 5G Core
TS	Timestamp
K_hmac_	Symmetric Key for HMAC Generation
E_PKCore_ (msg)	Encrypt Message with the Public Key of 5GC

**Table 4 sensors-22-04525-t004:** Experimental time for different functions.

Notation	Description	Time (ms)
$T_{ECDSA.sign}$	ECDSA Sign	0.7286
$T_{ECDSA.ver}$	ECDSA Verify	1.3442
$T_{ECIES.enc}$	ECIES Encryption	2.0572
$T_{ECIES.dec}$	ECIES Decryption	0.7851
$T_{HMAC}$	KDF/HMAC Calculation	0.0495
$T_{HMAC.ver}$	HMAC Verification	0.0281
$T_{ECDH.gen}$	ECDH Key Generation (1 Exp)	0.6945
$T_{ECDH.CK}$	ECDH Common Key	0.7099
$T_{hash}$	SHA256 Calculation Time (Hash Time)	0.0206
$T_{sym.enc}$	Symmetric Encryption	0.0925
$T_{xor}$	XOR Operation	0.0084
$T_{BC.read}$	Blockchain Transaction Read	0.2914
$T_{BC.write}$	Blockchain Transaction Write	0.0434
$T_{DB.read}$	Centralized Database Read	0.4956

**Table 5 sensors-22-04525-t005:** Theoretical computational overheads.

Protocol	Entity	Authentication Computational Overhead	Execution Time (ms)
5GSBA	UE	$T_{ECIES.enc}+T_{ECDH.gen}+T_{hash}+T_{HMAC}+T_{HMAC.ver}$	2.8499
	CN	$T_{ECIES.dec}+T_{ECDH.gen}+T_{hash}+T_{HMAC}+T_{HMAC.ver}+T_{BC.read}+T_{BC.write}$	1.9126
	Both	$T_{ECDH.CK}$	0.7099
5G-AKA	UE	$T_{ECIES.enc}+2T_{sym.enc}+T_{xor}+2T_{HMAC}$	2.3496
	CN	$T_{ECIES.dec}+2T_{sym.enc}+T_{xor}+2T_{HMAC}+2T_{hash}+T_{DB.read}$	1.6143
SE-AKA	UE	$T_{ECIES.enc}+4T_{HMAC}+T_{HMAC.ver}+T_{ECDH.gen}+T_{ECDH.CK}$	3.6877
	CN	$T_{ECIES.dec}+2T_{HMAC.ver}+3T_{HMAC}+T_{ECDH.gen}+T_{ECDH.CK}+T_{DB.read}$	2.8898
BB-AKA 5G	UE	$T_{ECDSA.sign}+2T_{ECDSA.ver}+T_{ECDH.gen}$	4.1115
	CN	$3T_{ECDSA.sign}+2T_{ECDSA.ver}+T_{ECDH.gen}+T_{BC.read}+T_{BC.write}$	5.9035
	Both	$T_{ECDH.CK}+T_{hash}$	0.7305

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
