# Peer review of "A Secure Blockchain-Based Authentication and Key Agreement Scheme for 3GPP 5G Networks"

_sensors, 2022, doi:10.3390/s22124525_

Round 1

Reviewer 1 Report

Although the 3GPP 5G standard has been applied to the real world, it is interesting to evaluate and propose new methods focusing on incrementing the security in this scenario.

The author proposes a blockchain-based protocol to enhance the security of 5G.

The template needs to be verified if it is correct for the journal.

All acronyms need to be defined before being used.

The paragraphs do not have an identation.

It is not clear how the revocation and updating will work.

It is necessary to increase the discussion about protocol's overhead.

The conclusion section should be improved.

Reviewer 2 Report

The subject of the article is interesting. Below are the issues that have been formulated after reading the article carefully.

- The charts in the drawings could be of a larger font;

- scection "Conclusion": could be extended to include specific research findings as this section now contains very general conclusions.

Reviewer 3 Report

The paper proposes a new approach that improves the robustness of the 5G networks using secure blockchain-based authentication.  

The state of the art is based on 32 reference titles, from which only 12 are from the last 3 years. Considering the dynamics of the domain, I consider that it would be beneficial for the authors to increase the number of very recent reference titles, in order to anchor their research in most actual research trends.

- regarding the 5GSBA architecture, please underline the improvements with respect to the existing architecture.

- also, since

- equations presented (steps) should be numbered, with the number aligned at the right margin of the paper, and all the actors involved (parameters, notations, etc) should be explained in the text. It is not clear in the equations from section V.A.1 what the operators represent and not all the parameters are available in the accompanying text. A list explaining the significance of the operators and another one explaining the meaning of all variables involved might be an asset.

- “cryptography assumption and the Dolev-Yao adversary model” [reference is needed],

- Figure 4 - Formal Verification with Scyther - brings no information for the reader – it can be explained I the text what the formal verification is showing.

- when comparing the results with other algorithms, namely 5G-AKA, SEKA and BB-AKA 5G reference are necessary for the existing algorithms.

- at row 578 “we derive the average delays in the execution of the protocols in Figure …” – it is not clear if it has been mathematically deduced, measured, evaluated using other means. Please explain.

- same for the results shown in figure 8. How you evaluated those results.

- the formulas presented in VI.C should be written as equations. Where have you taken the constants from (have they been measured? Taken from existing literature? Other?)

- How did you evaluated the results presented in Figure 10?

- The conclusion section should be extended to highlight once again the novelty of the approach, the improvement brought and open issues / further research ideeas.

At last, but not al least, the format required by MDPI has not beeb respected, Please check it once again.

Round 2

Reviewer 3 Report

The paper looks much better now and I consider that it can be published in a prestigious journal like Sensor.